# Technique and early results of endovenous laser ablation in morphologically complex varicose vein recurrence after small saphenous vein surgery

**Lars Müller** [1,2]*, **Eike Sebastian Debus**[2], **Syrus Karsai**[3,4], **Jens Alm**[1]

1 Department of Vascular Surgery, Dermatologikum Hamburg GmbH, Hamburg, Germany, 2 Department of Vascular Medicine, University Heart and Vascular Center, University Medical Center Hamburg-Eppendorf, Hamburg, Germany, 3 Department of Dermatology, Dermatologikum Hamburg GmbH, Hamburg, Germany, 4 Department of Dermatology, University Medical Center Hamburg-Eppendorf, Hamburg, Germany

* L.mueller@dermatologikum.de

**Data Availability Statement:** All relevant data are within the manuscript and its Supporting Information files.

## Abstract

### Background

Recurrences after varicose vein treatment occur frequently and represent a significant health and economic problem. In contrast to primary treatments of superficial truncal venous insufficiency, their management is often more difficult. Here we assessed the technical feasibility and early results of endovenous laser ablation (EVLA) for recurrences with stumps or subfascial meandering varices after small saphenous vein (SSV) surgery.

### Methods

This single-center retrospective study included 45 consecutive EVLA procedures from July 2019 to December 2021 in 40 patients (19 male, 21 female, mean age 62.8 ± 12.7 years). Patients had clinically relevant recurrent varicose veins after SSV surgery, with clinical, etiologic, anatomic and pathophysiologic (CEAP) classification stage of C2S or higher. We categorized the recurrences morphologically according to duplex sonographic criteria. Ablations were done with a 1470 nanometers laser and dual ring radial fibers and aimed at thermal closure as proximal as possible to the upper inflow. Analyses were performed by descriptive statistics and the Kaplan-Meier method. The primary outcome analyzed was the technical success, defined by thermal occlusion not requiring re-intervention during the observation period. The secondary outcome was the occurrence of postoperative complications.

### Results

A complex morphology with residual stumps or tortuous venous connections to the popliteal vein was present in 35 cases (77.8%). Immediate technical success at the first postoperative visit after a median of 11 days (interquartile range 8–13 days) was 97.8%. During the follow-up period (median 77 days, interquartile range 13–256 days), 6 limbs (13.3%) required redo EVLA due to symptomatic persistent or newly presenting reflux. The median freedom from re-recurrence was 791 days. Otherwise, no medical or surgical complications requiring

**Funding:** Parts of our author group (L.M. and S.K.) are employed by Dermatologikum Hamburg GmbH, a non-academic, commercial company. The employer and funder provided support in the form of salaries for these authors [L.M., S.K.], but did not have any additional role in the study design, data collection and analysis, decision to publish, or preparation of the manuscript. The specific roles of these authors are articulated in the 'author contributions' section.

**Competing interests:** Parts of our author group (L. M. and S.K.) are employed by Dermatologikum Hamburg GmbH, a non-academic, commercial company. L.M. received lecture fees and financial support for travel expenses and congress visits by Biolitec AG. The commercial affiliations displayed do not alter our adherence to PLOS ONE policies on sharing data and materials. There are no patents, products in development or marketed products associated with this research to declare.

specific treatment were observed, particularly no endothermal heat-induced thrombosis (EHIT) or other thrombotic complications, and no nerve damage.

## Conclusion

According to our pilot data, EVLA is technically feasible for complex popliteal variceal recurrence, although the success rate appears substantially lower than for primary treatment of truncal venous insufficiency.

## Introduction

Superficial venous incompetence is a common venous disorder that is often progressive and may lead to chronic venous insufficiency. According to the Clinical, Etiological, Anatomical, Pathophysiological classification (CEAP), and based on epidemiologic data, up to 21% of people in Europe suffer from varicose veins (C2) and up to 13% from chronic venous insufficiency (C3-C6) [1]. Surgical or interventional treatment of symptomatic superficial reflux (CEAP 2S) is considered cost-effective and beneficial in terms of quality of life. Endovenous laser ablation (EVLA) is preferred over surgery and foam sclerotherapy treating primary truncal vein insufficiency in contemporary practice guidelines [1].

Recurrent varicose veins can occur after any form of treatment. Causes include neovascularization, recanalization, disease progression and tactical or technical errors during pre-treatment [2]. The incidence of recurrence has been reported to be up to 25% after only 5 years and increases over time [3].

In contrast to primary treatment, no well-established treatment algorithms exist for recurrent varicose veins. In the case of persistent incompetent saphenous trunks, endovenous thermal treatment is recommended [1, 4]. When straight truncal veins are not present, such as in short-stretch vein segments or residual stumps, foam sclerotherapy or surgery are considered [1]. In contrast, there are insufficient study data to favor the use of thermal ablation techniques in these situations [1, 4, 5]. Technical progress, advanced by the medical device industry, is resulting increasingly in techniques that enable short vein sections to be ablated thermally. These include special radiofrequency catheters with a short heating segment, bare laser fibers, and laser fibers with radial energy emission [6, 7].

Recurrent varicose veins of the small saphenous vein (SSV) arising from the previous saphenopopliteal junction (SPJ) present a particular therapeutic challenge. Even in primary truncal venous insufficiency of the SSV, there is considerable variability with regard to the anatomy and the origin of the pathological reflux in the area of the confluence with the deep venous system [8]. The morphological appearance of recurrences is also heterogeneous [9], and there exist only few reports in the literature which describe endovenous thermal treatments of a residual SSV stump or recurrent tortuous venous segments arising in the popliteal fossa [6]. In such cases, the cannulation and ablation of the subfascially localized upper insufficiency point by thermal ablation is often a technical challenge—not only because of the meandering appearance of the recurrences, but also because of the presence of the nearby deep vein and tibial nerve. Incidentally, for the same reasons, open surgical revision of varicose vein recurrence in the popliteal fossa is considered a demanding procedure [10–13].

The aim of our pilot study was therefore to investigate the feasibility and efficacy of thermal ablation as a potentially less invasive procedure in such situations using our own cases from a specialized center.

## Methods

### Study design

This was a retrospective study. We retrospectively reviewed all cases of EVLA performed by one operator (LM) over a 30-month period, from July 2019—December 2021. Of these, all cases with recurrence treatment after SSV surgery were selected. The data were extracted from the electronic patient records from March 1, 2024, anonymized and stored in a fully anonymized database (S1 File) and accessed for research purposes. After the data collection, no information was accessible with which individual participants could be identified. The STROBE guidelines (Strengthening Reporting on Observational Studies in Epidemiology) were employed to review reporting in this study [14].

### Ethics statement

The conduct of the study is in accordance with the principles of the Declaration of Helsinki and has been approved by the responsible Ethics committee of the Hamburg Medical Association. Due to the retrospective study design which is based on anonymized data, the need to obtain informed consent was waived (file reference PV7252-4650_1WF).

### Preoperative diagnostics

The diagnoses were made with duplex ultrasound using a Logiq P6 Pro (GE Healthcare, Chicago, IL) device while the patient was standing. Endovenous laser therapy was indicated when hemodynamically and clinically relevant popliteal recurrence occurred after either operative or endothermal SSV surgery with Clinical, Etiological, Anatomical and Pathophysiological (CEAP) class C2S or higher. A minimum diameter of 3 mm and a reflux duration of > 0.5 seconds after manual calf compression were needed. In all instances, thermal closure was considered to be the most effective and gentle procedure due to the anatomy and extent of reflux flow. The ablation strategy was determined individually during duplex mapping. The aim was to obtain a thermal closure as proximally as possible, i.e. at the connection to the deep venous system. An important prerequisite here was that a vein segment of at least 1 cm in length was available into which the end of the radial laser fiber could be inserted in order to safely transfer the thermal energy into the vein wall. Distal vein segments were also thermally ablated if it was possible to insert a cannula and place the radial laser fiber over a reasonably long distance. A classification from the literature was used to categorize the existing variability in morphology [9]: cases with a largely intact SPJ and a preserved SSV (type 1); varices of the popliteal fossa communicating with the popliteal vein via a proximal short or tortuous stump, which corresponds to either a persistent SPJ or a prominent perforating vein (type 2); a remaining venous segment of the SSV communicating with the popliteal vein via tortuous veins like neovascularizations or perforators rather than the junction (type 3); and recurrences in which a distal refluxing segment of the SSV with no obvious communication with the popliteal vein remained, but which was considered sufficiently hemodynamically relevant and causative for the existing varicosities and venous symptoms (type 4). A schematic illustration of this morphological classification is given in Fig 1.

### Endovenous ablation technique

All procedures were performed under sterile conditions with ultrasound guidance (Logiq e, GE Healthcare, Chicago, IL) and the patient in the prone position.

Depending on the size of the fibers used, 4F indwelling vein cannulas were placed, or 6F venous sheaths were introduced according to the Seldinger method. For short residual stumps

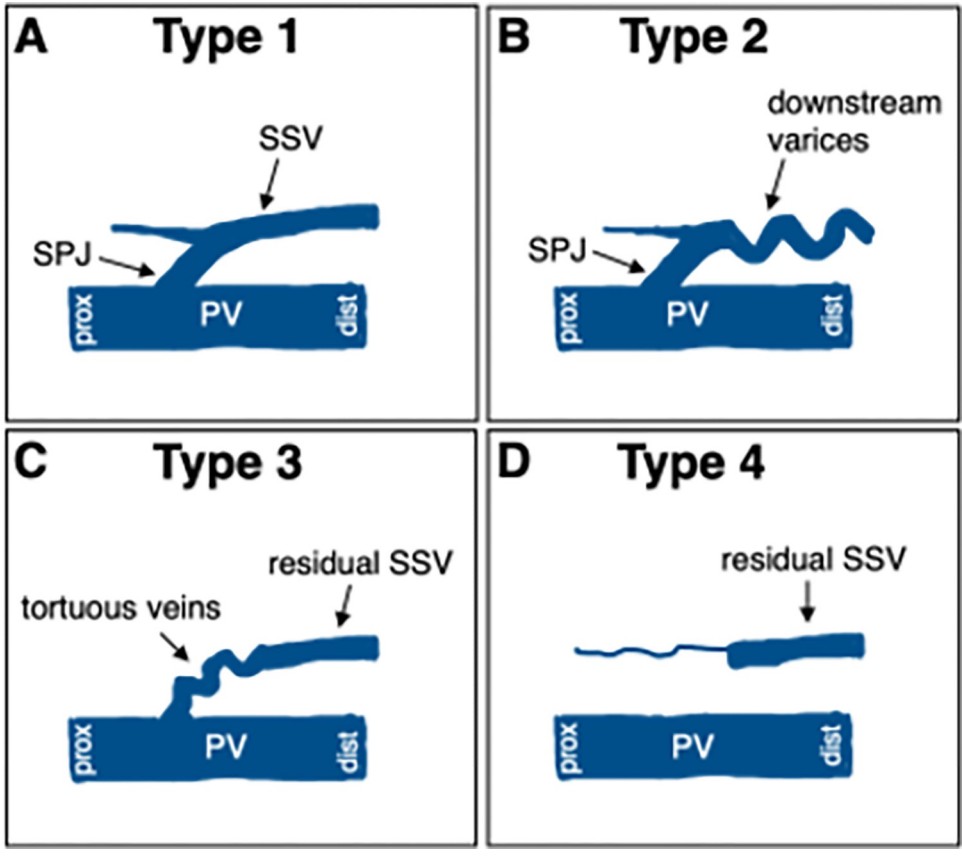

**Fig 1. Illustration explaining the different morphological types of SPJ recurrence according to Tong and Royle** [9]. A, type 1 representing a largely intact small saphenous vein with preserved saphenopopliteal junction. B, type 2 with varices communicating with a residual SPJ stump or prominent perforators. C, type 3 with remaining SSV segment connected proximally to tortuous, neovascularization-like veins. D, type 4 representing preserved SSV-segments without significant communication to the popliteal vein. PV: popliteal vein; SPJ: saphenopopliteal junction; SSV: small saphenous vein; prox: proximal; dist: distal. PV: popliteal vein.

or tortuous vein segments, the affected vein segments were directly punctured and cannulated (Fig 2A and 2B), whereas longer, straight truncal vein segments were treated from a distally located access. The fiber tip was placed directly at the source of reflux at the junction with the deep venous system. The aim was to achieve a largely flush closure (Fig 2C). However, we did

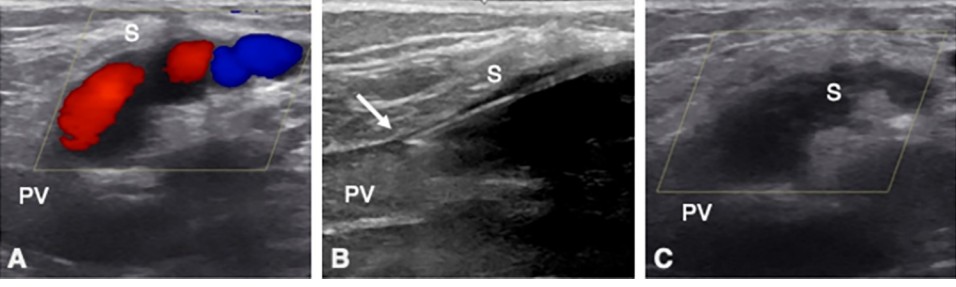

**Fig 2. Endovenous laser ablation of a SSV recurrence with a stump.** A, representative duplex ultrasound image showing a residual SPJ stump (S) preoperatively. B, during therapy, the puncture cannula (arrow) was advanced into the remaining stump under ultrasound guidance. C, postinterventional ultrasound showing thermal occlusion of the stump. PV: popliteal vein.

not thermally occlude muscle veins that joined the SSV to form a common trunk. If necessary, a sufficiently large distance to the SPJ was left here.

For particularly tortuous areas, the puncture canula was threaded inside the torturous vein using ultrasound guidance. For more complex findings, several venous cannulas were inserted into the sections to be sealed before infusion of the tumescence solution. Fig 3 illustrates this technique using a case study. The preparatory placement of the cannulas allows the most important vein sections to remain accessible even after injection of the tumescent solution, which means that only one laser fiber is required for the entire treatment. Ablation was performed using a 1470 nm diode laser with either the slim 1.3 mm or 1.9 mm dual-ring radial fibers (ELVeS® Radial®, Biolitec AG, Vienna, Austria). Before laser emission, tumescent anesthesia was administered (1000 ml physiological saline + 50 ml mepivacaine 1% + 8 ml sodium bicarbonate 8.4%). The tumescent solution is positioned around the vein to be treated like a cuff before the laser energy is released. Particular care was taken to achieve an adequate thermal barrier between the vein and the nerve structures in the popliteal fossa. The tibial nerve and, if necessary, the sciatic nerve and the common peroneal nerve were identified. Before the energy was applied, the tumescent solution was injected in a targeted manner between the vein and the nerve in order to achieve a distance of at least one centimeter. Varicose veins located directly under the skin were also treated, using targeted tumescent infusion to maintain a distance of at least one centimeter between the tip of the catheter and the surface of the skin. The power output was 8 W (1.3 mm fiber) or 10 W (1.9 mm fiber). The linear endovenous energy density (LEED) target was 50–100 joules per cm of vein segment, dependent of the vein diameter. As an approximation, it can be said that an energy output of about 10 J/cm was aimed for with each millimeter of vein diameter. Since there were often no uniform vein diameters over longer distances, it was considered helpful to achieve a gapless visual, ultrasonographic assessment of sufficient thermal response with respect to echoreflexes due to gas bubbles. Also, a certain haptic resistance, but no sticking, was aimed for when pulling back during firing. It should be noted that the aforementioned approaches are not scientifically established methods of controlling energy delivery, nor are they readily reproducible or measurable.

No simultaneous intraoperative applications of sclerosing agents were performed in any case. Also, no concomitant sclerotherapy or phlebectomy of tributaries was performed.

Patients were offered to have the treatments under light sedation with propofol and in the presence of an anesthesiologist. Sterile tape dressings, but not compression bandages, were administered postoperatively. Although there is no scientific evidence of efficacy, all patients received a prophylactic intraoperative injection of low-molecular-weight heparin. This is an institutional practice used in all endovenous laser ablation procedures. If the patient's medical history indicated an elevated risk of thrombosis, thromboprophylaxis was continued for up to seven days. Postoperatively, before discharge, the motor function was checked. All patients were given an emergency phone number postoperatively with availability 24/7 and asked to use it in case of problems.

## Follow-up examinations

The first postoperative examination was carried out within 14 days. The aim of the procedure was the sonographically detectable proximal occlusion and elimination of reflux (Fig 2C) and decompression of the downstream varicosities. A clinical evaluation was performed to exclude relevant complications. In this context, moderate bruising and tolerable pain, which may have been treated with analgesics, were not considered complications. The patients were also asked about motor or sensory deficits during the follow-up examination. The occlusion of

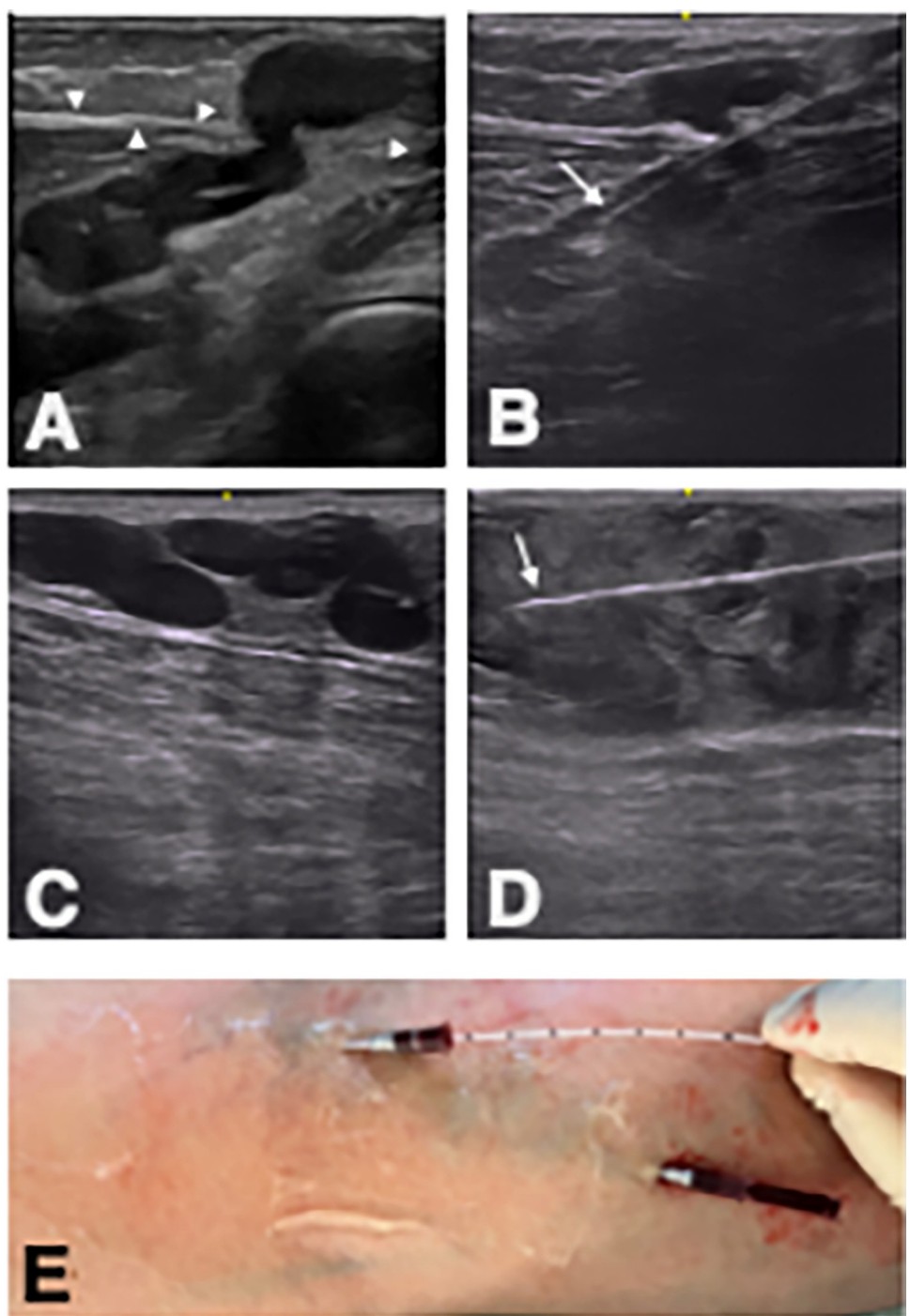

**Fig 3. Typical example of preparation of an EVLA in a patient with type 2 recurrence in the right leg.** A, ultrasound longitudinal section of the recurrence formation originating below the popliteal fascia and attached varicose portions also above the fascia (arrowheads = popliteal fascia). B, the proximal part is cannulated first (arrow). C, longitudinal ultrasound image of prominent, attached varices, which are located directly subcutaneously, above the fascia. D, longitudinal ultrasound image after cannulation and infusion of the tumescent solution. By infusing a substantial amount of tumescent solution, a sufficient distance of at least 1 cm to the skin surface is ensured despite the subcutaneous localization of the varicose veins. E, the inserted cannulas have a length of 5 cm and ensure that the previously secured vein sections can be ablated with one laser fiber despite the infusion of the tumescent solution.

downstream varices, which sometimes corresponded to the transient clinical presentation of a localized superficial vein thrombosis, was also not regarded as a complication here. Further evaluations were offered at intervals of 3–6 months.

Occasionally, persistent, superficial varices were treated with sclerotherapy with polidocanol during these postinterventional visits. The presence and treatment of such veins were not regarded as a treatment failure or re-recurrence. Re-Recurrence requiring redo intervention by EVLA was defined as clinically relevant, duplex sonographically detectable refluxing (>0.5 seconds) veins of 3 mm or more that were the result of recanalization or new formation at the pretreated site.

### Data analysis

The primary outcome measure was the success rate, i.e., thermal closure of recurrent varices that did not require further or repeat interventional treatment. The results of these redo interventions were not evaluated in the present study. The secondary outcome parameter was the postoperative morbidity.

To describe the baseline parameters, frequencies are reported for categorical variables. The metric data was analyzed for normal distribution using the Kolmogorov-Smirnov test. For categories which exhibited skewed data, we displayed the data with medians, ranges and interquartile ranges (IQR). Normally distributed data were presented with means and standard deviation (SD). When calculating the median follow-up time, all observations were taken into account, regardless of whether an event occurred or whether they were censored. For the Kaplan-Meier analysis and curve generation, GraphPad Prism (Version 9.5.1) was applied (GraphPad Software Inc, San Diego, CA, USA).

## Results

During the study period, a total of 1442 EVLA treatments were performed. Of these, forty patients with forty-five treated limbs for SSV recurrences were enrolled. Five patients had bilateral findings. Basic patient characteristics, intended to provide insight into the constitution of the cohort, and treatment characteristics are summarized in Table 1.

The SSV recurrences requiring treatment had very diverse morphology. An appropriate classification of recurrent morphological patterns was used to provide some characterization [9]. Cases in which a largely normally configured small saphenous vein with straight vein courses was recognizable despite previous surgery were sometimes observed. These constellations of findings, characterized as type 1 (n = 4; 8.9%), are rather less technically demanding and can mostly be instrumented coming from distal. However, more complex situations with short stumps (type 2; 55.6%) or residual SSV segments communicating with the popliteal vein via tortuous veins (type 3; 22.2%) predominated in the present cohort (Fig 4; Table 1). With types 2 and 3, it was generally not possible to position the laser fiber via a puncture from distal to the upper insufficiency point through a reasonably straight vein. In contrast, the multiple axial displacements require direct puncture of the most proximal part of the recurrence and additional cannulation and treatment in individual sections of the downstream veins. Fig 4 shows examples of different cases to illustrate this circumstance.

All treatments were scheduled and performed in a strictly outpatient setting, and no patient required monitoring for more than two hours afterwards.

Every treated patient came for a post-interventional control, which we recommended after 10–14 days. The median time until this first follow-up was actually performed was 11 days (range 1–61 days, IQR 8–13 days). Only one procedure was considered an early failure. Accordingly, the initial ablation success at the first postoperative follow-up was 97.8%. With a

**Table 1. Baseline characteristics of the patients and treatment sessions.**

| Characteristic | Value |
|---|---|
| *Patients* | |
| Number of patients | 40 |
| Sex; n (%) | |
| Male | 19 (47.5) |
| Female | 21 (52.5) |
| Age (years); mean (±SD) | 62.8 (±12.7) |
| Body mass index (kg/m2); mean (±SD) | 24.5 (±2.9)[a] |
| ASA score; n (%) | |
| ASA I | 16 (40) |
| ASA II | 22 (55) |
| ASA III | 2 (5) |
| *Treatments* | |
| Number of treatments | 45 |
| Side; n (%) | |
| Left | 27 (60) |
| Right | 18 (40) |
| Vein diameter (mm); median; range, IQR | 5; 4–10; 5–7 |
| Anatomic type; n (%) | |
| Type 1 | 4 (8.9) |
| Type 2 | 25 (55.6) |
| Type 3 | 10 (22.2) |
| Type 4 | 6 (13.3) |
| Previous intervention; n (%) | |
| Endovascular | 13 (28.9) |
| Surgical | 32 (71.1) |
| Energy output (J); mean (±SD) | 519.1 (±356.4) |
| Treated vein length (cm); mean (±SD) | 8.2 (±6.2) |
| LEED (J/cm); mean (±SD) | 70.4 (±24) |
| Laser fiber; n (%) | |
| 1470 nm dual ring 1.9 mm (10 W) | 6 (13.3) |
| 1470 nm dual ring 1.3 mm (8 W) | 39 (86.7) |
| Administration of sedatives; n (%) | |
| Yes | 20 (44.4) |
| No | 25 (55.6) |

ASA: American Society of Anesthesiologists; SD: standard deviation; IQR: interquartile range; LEED: Average linear endovenous energy density

[a]Data missing in 4 patients

median observation period of 77 days (range 1–920 days; IQR 13–256 days), the success rate, defined as thermal closure requiring no further reintervention in the treated area, was 86.7%. Overall, recurrence at the same site occurred in 6 cases (13.3%), requiring reintervention using EVLA. Table 2 further specifies the cases with re-recurrence. In three cases, re-recurrences were seen during follow-up, after the first postoperative visit. In two other cases, a satisfactory ablation result was present at the stump but with open and refluxing truncal vein remnants via other feeding veins distally.

The Kaplan-Meier curve (Fig 5) also indicates the observation time without recurrence and displays the censored observations as well. As can be seen from the diagram, many patients

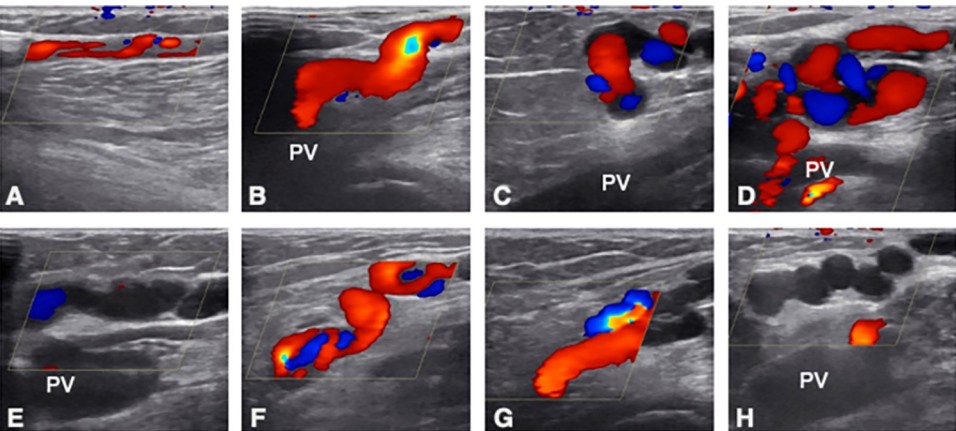

**Fig 4. Representative color duplex ultrasound images of different cases, longitudinal sectioning.** A, a recanalized truncal vein segment after pretreatment with thermal procedure, classified as type 1. B, C, different patients with typical stump and attached veins, categorized as type 2. D, massive finding with partial large caliber popliteal varicose veins, classified as type 2. E, almost straight vein segments without a stump clearly delineated, this finding was assigned as type 3. F, G, popliteal recurrences with stump (type 2), H, meandering, Neovascularization-like recurrence, without clearly sonographically definable stump (type 3). PV: popliteal vein.

did not participate in any further follow-up visits after the first post-intervention control. This is likely to result in some bias. The median recurrence-free time was calculated to be 791 days. A detailed time-to-event profile listing all cases and indicating time to recurrence or observation period without recurrence can be found in S1 Fig.

In 12 cases (26.7%), foam sclerotherapy of superficial varicosities with polidocanol was performed following EVLA during the follow-up period. These treated side branch veins appeared as residual varices with otherwise sufficient closure of the recurrence in the popliteal fossa. No further complications or medical problems requiring drugs or interventions occurred after treatment. Additionally, we did not observe any thromboembolic events, such as deep venous thrombosis, endothermal heat-induced thrombosis (EHIT), or pulmonary embolism. Moreover, there was no motor or sensory nerve damage and no thermal skin damage.

## Discussion

The treatment of recurrent varices, in contrast to the treatment of incompetent truncal veins, is challenging. Open revision surgery for popliteal varicose vein recurrence is associated with some risk of complications, such as wound infection, sensory and motor nerve lesions,

**Table 2. List of cases with inadequate treatment outcomes and indications for reintervention during the follow-up period.**

| No. | Sex | Side | Morph. type | Max. diameter (cm) | Energy output (J) | Time to indication for repeat procedure (days) | Failure reason | Rating |
|---|---|---|---|---|---|---|---|---|
| 1 | male | right | 2 | 7 | 1204 | 441 | Residual open distal vein segment, the proximal stump was sufficiently closed. | Partial success |
| 2 | female | left | 3 | 5 | 627 | 791 | Newly emerged refluxing stump | Recurrence |
| 3 | female | left | 3 | 6 | 364 | 517 | Recanalization | Recurrence |
| 4 | male | right | 2 | 9 | 259 | 11 | Inadequate ablation result, very complex condition | Failure |
| 5 | male | left | 2 | 10 | 410 | 112 | Residual vein segments distally, proximal stump adequately occluded | Partial success |
| 6 | male | left | 2 | 7 | 175 | 920 | Newly emerged refluxing stump | Recurrence |

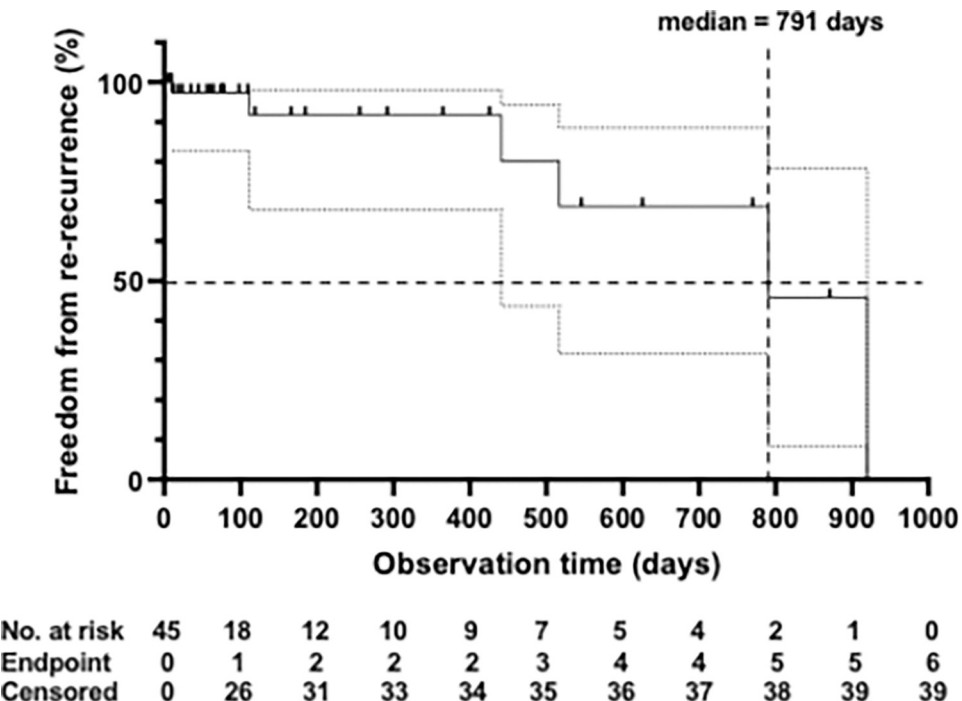

**Fig 5. Kaplan-Meier curve indicating the technical success of EVLA for popliteal recurrence.** The occurrence of a re-recurrence was considered an event. The 95% confidence intervals are indicated. The number of cases at risk, events and those censored are displayed below the time axis.

lymphatic fistulae, edema, and deep venous thrombosis [10–13]. In this respect, endovenous thermal procedures offer potential advantages as they are performed on an outpatient basis and are potentially associated with lower morbidity. Apart from the treatment failures no relevant peri-interventional side effects were observed in the cohort studied here. Whether an increased morbidity is to be expected due to the fact that recurrences are treated here cannot be adequately assessed due to the small number of cases. However, the requirements and preparations for recurrence treatment do not differ from primary treatment and correspond to general basic surgical principles: asepsis, preparation in advance, adequate intraoperative imaging, knowledge of anatomy, and protection of the anatomical structures. The success rate in our present observation is inferior to the good outcome of EVLA in the treatment of SSV recurrence with preserved venous trunk [4]. A more recent study investigated the success of thermal ablation in redo treatments of SSV recanalization and reported a success rate of 96.5% with a mean follow-up of 24.9 ± 14.9 months [15]. This difference can probably be explained by the anatomically more complex recurrence characteristics in our present cohort. The effectiveness of thermal energy transfer to the vein wall is presumably lower when only short segments are ablated as opposed to longer, straight trunk vein segments.

In addition to thermal ablation, ultrasound-guided foam sclerotherapy would be an effective, even less invasive, easier-to-handle procedure [16, 17]. Current guidelines consider sclerotherapy to be the most suitable treatment for recurrences without residual truncal vein [1]. However, we deliberately opted for the thermal ablation procedure here, as sclerotherapy was not considered very promising due to the vein diameters and the extent of the reflux. In primary treatment of SSV incompetence, the efficacy of sclerotherapy is regarded to be lower than that of thermal ablation or surgical techniques [18]. In contrast to the injection of the sclerosing agent, thermal ablation can be controlled more precisely. With sclerotherapy in the

veins of the popliteal fossa, however, there is a concern that the sclerosing agent can accumulate to a not inconsiderable extent in the deep veins, which poses a risk of thrombosis. For the same reason, in contrast to a report from an Italian case series study [6], we did not perform sclerotherapy as a concomitant procedure. Nevertheless, the results presented should not be interpreted as a claim that endovenous laser ablation is superior to sclerotherapy in the case of recurrence, as there are no corresponding studies to date.

It would also be conceivable to combine foam sclerotherapy with thermal ablation techniques. Especially in more widespread varices, complete regression after thermal ablation of a feeding stump or truncal vein remnant does not always occur, and the combination of laser ablation with sclerotherapy in one session may potentially reduce the need for further sessions. On the other hand, a successful ablation of the proximal part of a recurrence is frequently associated with a remarkable decompression of downstream varicosities and relief of local venous symptoms. It is still controversial whether simultaneous or delayed treatment of these varicose veins is beneficial [19–21]. In our routine for SSV recurrences, we tend to allow the superficial tributaries to collapse subsequently and offer treatment of the remaining varices after 3 to 6 months. The existing but decompressed varicose veins are merely a cosmetic nuisance and further regression of such branches often occurs. Subsequent sclerotherapy of such veins is often only a minor procedure.

In the case of tortuosity at the proximal end of the recurrence, as represented by morphologic type 3, the simultaneous application of foam, for example via an additional catheter lumen, could also have been a useful adjunct [22]. Even more extensive use of thermal treatment of subcutaneously localized varicose veins [23] could possibly have reduced the frequency of sclerotherapy (26.7% in this case series) and will be used more frequently in the future.

The radially emitting fibers used for the treatment enable the ablation of very short vein sections. However, it must be assumed that other devices, such as bare laser fibers or radiofrequency catheters, which can also emit targeted heat over a short distance, are also potentially suitable. Special techniques are repeatedly used in the treatment of recurrences and can offer improved results. First, the puncture technique can be applied in such a way that even tortuous vein segments are cannulated to the maximum extent and can thus be effectively thermally ablated. Second, for more complex findings, several venous cannulas were inserted into the sections to be sealed, thus ensuring ablation later with one catheter, even after infusion of the tumescent anesthesia. This technique is similar to that described as the hedgehog technique [24]. Ultimately, multiple sessions may be acceptable in the case of extensive findings, similar to what is common practice in foam sclerotherapy. Consequently, we could interpret the two cases in which the proximal inflow was blocked and reflux was still seen distally as at least partial success of the treatment (Table 2).

## Limitations

As the study only comprised 45 EVLA treatments, this is a rather small sample, which may affect the validity of the statistical results and their transferability. In addition to the small number of cases, the frequency of only six events of re-recurrence during follow-up does not provide a sufficient basis for subgroup analyses, let alone multivariate survival analyses. Nevertheless, in view of the reports published to date, this number is not insignificant, as there is actually hardly any comparable published data.

The median follow-up period of 77 days is inherently insufficient to assess the long-term effects and potential long-term complications of EVLA. Several patients did not participate in further examinations after the first follow-up. However, complete follow-up outside of

prospective studies is not easy to realize as part of the routine treatment of outpatients with varicose veins. Retrospective data can nevertheless be useful for planning prospective studies. It must also be mentioned that the report represents only the experience of a single surgeon from a single center. This can influence the results through biased selection. On the other hand, this setting results in a high degree of standardization and potentially a low information bias. Further studies investigating comparability and cost-effectiveness are necessary. To this end, it would be essential to validate methods that quantify reflux or the overall severity of recurrence, for example using duplex sonographic or radiological imaging techniques. This could serve as a basis for a comparison of the different approaches, e.g. foam sclerotherapy and thermal ablation, in prospective studies. Possibly a threshold value, if such a value exists, can be determined for the extent of the findings above which EVLA is superior to foam sclerotherapy. Another option for evaluating data from routine treatment would be to analyze data from registry studies. However, a general consensus regarding the characterization and quantification of recurrences would also be useful here. In this regard, the present pilot study may provide a realistic impression of peri-procedural morbidity and early outcomes, and a proposal of morphological characterization and could serve as a starting point for further studies.

## Conclusion

Our early results of EVLA for complex recurrent popliteal varicose veins are fairly acceptable but not comparable to those of EVLA as primary treatment. With sufficient application experience, EVLA using radial laser fibers appears feasible for more complex recurrences with low periprocedural morbidity. Possibly, the technique is a reasonable adjunct to already widely used sclerotherapy techniques. Whether and in which situations it may be superior to sclerotherapy in terms of clinical efficacy and cost-effectiveness must be clarified in further studies.

## Supporting information

**S1 Fig. Time-to-event profile, related to the occurrence of re-recurrences for which the indication for redo intervention using EVLA was given.** Cases are grouped within the four different morphological types and color-coded. Gray bars are cases with indication for reintervention. The morphologic type and observation time to event or censoring time are indicated next to the bars.
(TIF)

**S1 File. Raw data of the study.**
(XLSX)

## Author Contributions

**Conceptualization:** Lars Müller, Jens Alm.

**Data curation:** Lars Müller.

**Formal analysis:** Lars Müller, Eike Sebastian Debus, Jens Alm.

**Investigation:** Lars Müller, Eike Sebastian Debus, Syrus Karsai.

**Methodology:** Lars Müller, Syrus Karsai.

**Project administration:** Lars Müller.

**Supervision:** Lars Müller.

**Validation:** Lars Müller, Eike Sebastian Debus.

**Visualization:** Lars Müller, Syrus Karsai.

**Writing – original draft:** Lars Müller.

**Writing – review & editing:** Lars Müller, Eike Sebastian Debus, Syrus Karsai, Jens Alm.

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
