## [Decision Letter · Decision Letter 0]

28 Jun 2024

PONE-D-24-21826Technique and early results of endovenous laser ablation in morphologically complex varicose vein recurrence after small saphenous vein surgeryPLOS ONE

Dear Dr. Müller,

Thank you for submitting your manuscript to PLOS ONE. After careful consideration, we feel that it has merit but does not fully meet PLOS ONE’s publication criteria as it currently stands. Therefore, we invite you to submit a revised version of the manuscript that addresses the points raised during the review process.

We look forward to receiving your revised manuscript.

Kind regards,

Dr Redoy Ranjan, MBBS, MRCSEd, Ch.M., MS (CV&TS), FACS

Academic Editor

PLOS ONE

Journal Requirements:

"Parts of our author group (L.M. and S.K.) are employed by Dermatologikum Hamburg GmbH, a non-academic, commercial company. This does not change our adherence to the PLOS ONE guidelines for sharing data and materials. 

L.M. received lecture fees and financial support for travel expenses and congress visits by Biolitec AG."

We note that one or more of the authors are employed by a commercial company: Dermatologikum Hamburg GmbH

2) Please also provide an updated Competing Interests Statement declaring this commercial affiliation along with any other relevant declarations relating to employment, consultancy, patents, products in development, or marketed products, etc.  

Within your Competing Interests Statement, please confirm that this commercial affiliation does not alter your adherence to all PLOS ONE policies on sharing data and materials by including the following statement: ""This does not alter our adherence to  PLOS ONE policies on sharing data and materials.” (as detailed online in our guide for authors http://journals.plos.org/plosone/s/competing-interests). 

If this adherence statement is not accurate and  there are restrictions on sharing of data and/or materials, please state these. Please note that we cannot proceed with consideration of your article until this information has been declared.

3. Please upload a new copy of Figures 2, 3, and 4 as the detail is not clear. Please follow the link for more information:

https://blogs.plos.org/plos/2019/06/looking-good-tips-for-creating-your-plos-figures-graphics/

https://blogs.plos.org/plos/2019/06/looking-good-tips-for-creating-your-plos-figures-graphics/

Reviewers' comments:

Reviewer's Responses to Questions

**Comments to the Author**

1. Is the manuscript technically sound, and do the data support the conclusions?

Reviewer #1: No

Reviewer #2: Yes

Reviewer #3: Partly

2. Has the statistical analysis been performed appropriately and rigorously? 

Reviewer #1: Yes

Reviewer #2: Yes

Reviewer #3: Yes

3. Have the authors made all data underlying the findings in their manuscript fully available?

Reviewer #1: Yes

Reviewer #2: Yes

Reviewer #3: Yes

4. Is the manuscript presented in an intelligible fashion and written in standard English?

Reviewer #1: Yes

Reviewer #2: Yes

Reviewer #3: Yes

5. Review Comments to the Author

Reviewer #1: The draft is describing the method for treating a recurrent SSV, but unfortunately , this method is not reproducible and is not cost-effective for for majority of patients. However, it merits wide exposure for the specialists i the field.

Reviewer #2: Dear authors

It is a great privilege to review your article titled: Technique and early results of endovenous laser ablation in morphologically complex varicose vein recurrence after small saphenous vein surgery

Below please find comments, questions and suggestions

Title: OK

Abstract. Background: Please briefly describe the clinical setting and current status of venous disease. This is not a specialized journal and this could help readers

Methods: Remove case series, just leave retrospective study

Remove N

Add briefly your statistical tests in this section.

Spell out nm when using for the fist time in your abstract

Remove many ande leave the absolute or exact number (35)

Introduction.

Overall well-written, although I suggest to situate more in context in regards venous disease, stages and CEAP.

Methods: There is no need to leave Case series, please remove.

Figure 1, label A-D, indicates SPJ and vessel

Were all procedures bilateral?

Figure 3. From A to D is hard to visualize. Perhaps separating the E could help.

Figure 5, use arrows.

Results: Very well presented and described.

Discussion: This is often regarded as the most important section of the manuscript, where authors complete a deep analysis of their findings, compare with previous reports, point out learning strategies and propose further direction.

Overall well written.

Reviewer #3: This study evaluates the technical feasibility and early outcomes of endovenous laser ablation (EVLA) for complex varicose vein recurrences post-small saphenous vein surgery. The retrospective case series of 45 EVLA procedures showed a 97.8% initial success rate, with 13.3% requiring re-intervention during a median 77-day follow-up. The study concludes that EVLA is feasible for these complex cases, albeit with a lower success rate than primary treatments, and without significant complications. But there are still some limitations:

1. As this is a single-center retrospective study, its findings may be influenced by selection bias and the experience of a single surgeon. The study included only 45 EVLA procedures, a relatively small sample size that might affect the power of the statistical results and their generalizability.

2. The median follow-up time of 77 days might not be sufficient to assess the long-term effects and potential long-term complications of EVLA. Some patients did not attend further examinations after the first follow-up, which could lead to incomplete follow-up data, affecting the accuracy of the results.

3. I would suggest conducting a multifactorial analysis to assess other variables that may affect the results, such as patient characteristics, lifestyle, and comorbidities.

4. A more in-depth discussion in the discussion section may enhance the manuscript, including how the results compare with existing literature, as well as potential limitations and future research directions.

6. PLOS authors have the option to publish the peer review history of their article (what does this mean?). If published, this will include your full peer review and any attached files.

Reviewer #1: No

Reviewer #2: **Yes: **Carlos A. Hinojosa

Reviewer #3: No

---

## [Author Response · Author response to Decision Letter 0]

4 Jul 2024

1.PLOS ONE's style requirements, including those for file naming. 

-We have checked and adapted these style requirements

2. Funding Statement 

- The correct funding statement is:

Parts of our author group (L.M. and S.K.) are employed by Dermatologikum Hamburg GmbH, a non-academic, commercial company. The employer and funder provided support in the form of salaries for these authors [L.M., S.K.], but did not have any additional role in the study design, data collection and analysis, decision to publish, or preparation of the manuscript. The specific roles of these authors are articulated in the ‘author contributions’ section.

2.Competing Interests Statement

- Here is the adapted Competing Interests Statement:

Parts of our author group (L.M. and S.K.) are employed by Dermatologikum Hamburg GmbH, a non-academic, commercial company. 

L.M. received lecture fees and financial support for travel expenses and congress visits by Biolitec AG

The commercial affiliations displayed do not alter our adherence to PLOS ONE policies on sharing data and materials.

3. Please upload a new copy of Figures 2, 3, and 4 as the detail is not clear.

-We have reworked these images using the PACE software and hope that the quality is now OK.

Review Comments to the Author

Reviewer #1: The draft is describing the method for treating a recurrent SSV, but unfortunately , this method is not reproducible and is not cost-effective for majority of patients. However, it merits wide exposure for the specialists in the field.

- Thank you for your comment. We agree that reproducibility and cost-effectiveness is something that was not mentioned in the previous manuscript. We have made further comments on this in the Limitations section. We have also mentioned the need for clinical efficacy and cost-effectiveness in the Conclusions.

Reviewer #2:

Title: OK

Abstract. Background: Please briefly describe the clinical setting and current status of venous disease. This is not a specialized journal and this could help readers

- The abstract was expanded accordingly at the beginning under Background.

Methods: Remove case series, just leave retrospective study

Remove N

Add briefly your statistical tests in this section.

Spell out nm when using for the fist time in your abstract

Remove many and leave the absolute or exact number (35)

-Thank you for these suggestions for changes to the abstract, which we have tried to implement as best as possible.

Introduction.

Overall well-written, although I suggest to situate more in context in regards venous disease, stages and CEAP.

-We have included more background on superficial venous insufficiency at the beginning of the introduction. In addition, two further literature references have been added to the manuscript.

Methods: There is no need to leave Case series, please remove.

-We have removed this.

Figure 1, label A-D, indicates SPJ and vessel. Were all procedures bilateral?

-We have adapted Figure 1 accordingly. We have also added additional labels for the structures and the vessel direction (prox-dist). The caption for Figure 1 has also been expanded accordingly.

Figure 3. From A to D is hard to visualize. Perhaps separating the E could help.

-We hope that by re-uploading the image using PACE we will achieve improved quality here.

Figure 5, use arrows.

-We did not understand this comment exactly. We think the necessary information is included in the Kaplan-Meier curve or displayed below the curve. We do not know where the arrows should be placed here.

Results: Very well presented and described.

Discussion: This is often regarded as the most important section of the manuscript, where authors complete a deep analysis of their findings, compare with previous reports, point out learning strategies and propose further direction.

Overall well written.

-Thank you for your evaluation.

Reviewer #3: 

This study evaluates the technical feasibility and early outcomes of endovenous laser ablation (EVLA) for complex varicose vein recurrences post-small saphenous vein surgery. The retrospective case series of 45 EVLA procedures showed a 97.8% initial success rate, with 13.3% requiring re-intervention during a median 77-day follow-up. The study concludes that EVLA is feasible for these complex cases, albeit with a lower success rate than primary treatments, and without significant complications. But there are still some limitations:

1. As this is a single-center retrospective study, its findings may be influenced by selection bias and the experience of a single surgeon. The study included only 45 EVLA procedures, a relatively small sample size that might affect the power of the statistical results and their generalizability.

-We agree with this assessment and have explained this point under Limitations in the revision. Even if the statistical power is limited by a small number of cases, we currently have no better data in the literature. The data situation would improve if practitioners were to publish their undoubted experience.

2. The median follow-up time of 77 days might not be sufficient to assess the long-term effects and potential long-term complications of EVLA. Some patients did not attend further examinations after the first follow-up, which could lead to incomplete follow-up data, affecting the accuracy of the results.

-We pointed out, also under Limitations, that no conclusions on long-term results can be drawn from the data:

The median follow-up period of 77 days is inherently insufficient to assess the long-term effects and potential long-term complications of EVLA. Several patients did not participate in further examinations after the first follow-up. However, complete follow-up outside of prospective studies is not easy to realize as part of the routine treatment of outpatients with varicose veins. Retrospective data can nevertheless be useful for planning prospective studies. It must also be mentioned that the report represents only the experience of a single surgeon from a single center. This can influence the results through biased selection. On the other hand, this setting results in a high degree of standardization and potentially a low information bias.

3. I would suggest conducting a multifactorial analysis to assess other variables that may affect the results, such as patient characteristics, lifestyle, and comorbidities.

- For subgroup analyses, even with regard to univariate analyses, the number of cases and observations appeared too low. We therefore deliberately left all cases in one cohort. 

The number of cases and observations also appeared too small for possible multivariate analyses, such as Cox proportional hazard or multifactor analysis.

We have also commented on this under Limitations:

In addition to the small number of cases, the frequency of only six events of re-recurrence during follow-up does not provide a sufficient basis for subgroup analyses, let alone multivariate survival analyses.

4. A more in-depth discussion in the discussion section may enhance the manuscript, including how the results compare with existing literature, as well as potential limitations and future research directions.

-Thank you for the suggestion. As a result, we have significantly expanded the entire paragraph on Limitations, also with regard to the previous comments. 

In the discussion, a further technical aspect and a corresponding reference (Ref. 24) were added.

Overall, very few comparable studies or cases have been published to date, as far as is known. To the best of our knowledge, these have been taken into account in the Introduction and Discussion.

The following comments on possible further studies has also been added in the Limitations section:

Further studies investigating comparability and cost-effectiveness are necessary. To this end, it would be essential to validate methods that quantify reflux or the overall severity of recurrence, for example using duplex sonographic or radiological imaging techniques. This could serve as a basis for a comparison of the different approaches, e.g. foam sclerotherapy and thermal ablation, in prospective studies. Possibly a threshold value, if such a value exists, can be determined for the extent of the findings above which EVLA is superior to foam sclerotherapy. Another option for evaluating data from routine treatment would be to analyze data from registry studies. However, a general consensus regarding the characterization and quantification of recurrences would also be useful here.

---

## [Decision Letter · Decision Letter 1]

2 Aug 2024

PONE-D-24-21826R1Technique and early results of endovenous laser ablation in morphologically complex varicose vein recurrence after small saphenous vein surgeryPLOS ONE

Dear Dr. Müller,

Thank you for submitting your manuscript to PLOS ONE. After careful consideration, we feel that it has merit but does not fully meet PLOS ONE’s publication criteria as it currently stands. Therefore, we invite you to submit a revised version of the manuscript that addresses the points raised during the review process.

**ACADEMIC EDITOR: **The authors are thanked for this submission to the PLOS ONE. After a critical external peer review by experts and considering the overall reviewers' comments, I found that the paper has potential but requires improvements in clarity and presentation based on the reviewers' concerns. Please see the attached reviewer comments detail below.

We look forward to receiving your revised manuscript.

Kind regards,

Dr Redoy Ranjan, MBBS, MRCSEd, Ch.M., MS (CV&TS), FACS

Academic Editor

PLOS ONE

Journal Requirements:

Reviewers' comments:

Reviewer's Responses to Questions

**Comments to the Author**

1. If the authors have adequately addressed your comments raised in a previous round of review and you feel that this manuscript is now acceptable for publication, you may indicate that here to bypass the “Comments to the Author” section, enter your conflict of interest statement in the “Confidential to Editor” section, and submit your "Accept" recommendation.

Reviewer #1: All comments have been addressed

Reviewer #2: All comments have been addressed

Reviewer #3: All comments have been addressed

2. Is the manuscript technically sound, and do the data support the conclusions?

Reviewer #1: Yes

Reviewer #2: Yes

Reviewer #3: Yes

3. Has the statistical analysis been performed appropriately and rigorously? 

Reviewer #1: Yes

Reviewer #2: Yes

Reviewer #3: Yes

4. Have the authors made all data underlying the findings in their manuscript fully available?

Reviewer #1: Yes

Reviewer #2: Yes

Reviewer #3: Yes

5. Is the manuscript presented in an intelligible fashion and written in standard English?

Reviewer #1: Yes

Reviewer #2: Yes

Reviewer #3: Yes

6. Review Comments to the Author

Reviewer #1: (No Response)

Reviewer #2: Authors have addressed well my initial concerns.

Minor observation and important. Please add in you abstract, the number of patients, mean age, SD and gender proportions.

Reviewer #3: The study was inherently limited by the relatively small volume and short follow-up. The authors addressed my questions. No further comments.

7. PLOS authors have the option to publish the peer review history of their article (what does this mean?). If published, this will include your full peer review and any attached files.

Reviewer #1: **Yes: **Aram Baram

Reviewer #2: No

Reviewer #3: No

---

## [Author Response · Author response to Decision Letter 1]

12 Aug 2024

Dear Dr. Ranjan,

In the following, we would like to address the individual comments made by you and the reviewers.

We have prefaced our comments with each of your comments or points for consideration in bold italics.

Journal Requirements:

-We have looked through all the cited publications and can see that none of the papers have been withdrawn as of today's access date (August 12, 2024). However, we have replaced reference 14 with another article.

Review Comments to the Author:

Reviewer #2: Authors have addressed well my initial concerns.

Minor observation and important. Please add in you abstract, the number of patients, mean age, SD and gender proportions.

- We have included this information in the abstract.

---

## [Decision Letter · Decision Letter 2]

27 Aug 2024

Technique and early results of endovenous laser ablation in morphologically complex varicose vein recurrence after small saphenous vein surgery

PONE-D-24-21826R2

Dear Dr. Müller,

We’re pleased to inform you that your manuscript has been judged scientifically suitable for publication and will be formally accepted for publication once it meets all outstanding technical requirements.

Kind regards,

Redoy Ranjan, MBBS, MRCSEd, Ch.M., MS (CV&TS), FACS

Academic Editor

PLOS ONE

Review Comments to the Author

Reviewer #1: It is nicely organized and merits wide exposure to the international audience. It contains reproducible idea.

Reviewer #2: Dear authors.

Title Technique and early results of endovenous laser ablation in morphologically complex varicose vein recurrence after small saphenous vein surgery

Much improved manuscript. The initial concerns have been addressed well and your paper could be considered if there is agreement from all reviewers.

---

## [Editor Report · Acceptance letter]

3 Oct 2024

PONE-D-24-21826R2 

PLOS ONE

Dear Dr. Müller, 

I'm pleased to inform you that your manuscript has been deemed suitable for publication in PLOS ONE. Congratulations! Your manuscript is now being handed over to our production team.

Kind regards, 

on behalf of

Dr. Redoy Ranjan 

Academic Editor

PLOS ONE